# GKD-Recruiter: Jointly Modeling Social and Task Heterogeneity for Spatial Crowdsourcing via Graph Knowledge Distillation

Yucen Gao [1]  Zhemeng Yu [2]  Zhuoran Li [2]  Jianxiong Guo [3]  Xiaofeng Gao* [2]

## Abstract

Social recruitment offers a solution to worker scarcity in Spatial Crowdsourcing (SC) but faces challenges that are often ignored in traditional Influence Maximization. First, task heterogeneity arising from offline execution constraints breaks the "interest-implies-participation" assumption, as social influence often fails to translate into physical presence. Second, finite task demand creates a "saturation trap", a non-submodular setting in which utility drops sharply to zero once demand is met. To bridge these gaps, we propose GKD-Recruiter, a Task-Aware framework designed to maximize Effective Task Satisfaction (ETS). We explicitly model the complex worker-task affinity via a heterogeneous graph and capture directional social influence using a novel Influential GAT. To robustly fuse these distinct signals, we introduce a Graph Knowledge Distillation mechanism. Furthermore, we employ Rainbow DQN to navigate the non-submodular combinatorial search space, avoiding the local optima that trap greedy heuristics. Extensive experiments on real-world datasets demonstrate that GKD-Recruiter significantly outperforms state-of-the-art baselines in both solution quality and inference efficiency. The code is available at https://github.com/GaoYucen/GKD-Recruiter.

## 1. Introduction

Spatial Crowdsourcing (SC) has revolutionized location-based services, powering platforms like Uber, Didi Chuxing, and Meituan (Zhang et al., 2021; 2024). Despite their suc-

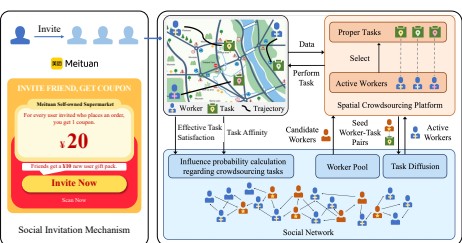

Figure 1. Motivation: (Left) Social recruitment mechanisms in real-world SC apps (e.g., referral bonuses, social sharing). (Right) Our modeling captures both social influence and task-specific affinity.

cess, these platforms face a persistent bottleneck due to worker scarcity, particularly in remote areas or during the "cold start" phase of new platforms. Traditional recruitment methods, which rely solely on the existing worker pool, often fail to meet the substantial and dynamic demand for spatial tasks (Gao et al., 2022). This limitation necessitates a paradigm shift toward leveraging the underlying social structure, thereby framing worker recruitment as an Influence Maximization (IM) problem in social networks.

Modern SC platforms are increasingly integrating social referral mechanisms to leverage the word-of-mouth effect (Zhan et al., 2024). For instance, in the gig economy, ride-sharing and delivery apps such as Uber and Meituan widely adopt invite-a-friend schemes, offering cash bonuses to existing workers who successfully recruit new riders from their social circles. Similarly, the emerging community group buying model, exemplified by platforms like Meituan Select and Duoduo Maicai, relies heavily on community leaders, often neighborhood store owners, to disseminate task information to neighbors via local social groups (e.g., WeChat groups). This mechanism effectively converts social ties into logistic distribution channels. Furthermore, location-based games like Pokémon GO encourage users to form social teams to complete co-located tasks. These real-world scenarios transform passive waiting among workers into an active information-diffusion process. Since budget constraints prevent platforms from incentivizing every user, it shifts to identifying a small subset of influential users, known as seeds, to trigger a cascade of participation. As illustrated in Figure 1, this naturally models the recruitment problem as IM on a social network (Li et al., 2023).

[1]Software College, Northeastern University, Shenyang, China [2]Shanghai Key Laboratory of Scalable Computing and Systems, School of Computer Science, Shanghai Jiao Tong University, Shanghai, China [3]Beijing Normal University, Zhuhai, China. Correspondence to: Xiaofeng Gao <gao-xf@cs.sjtu.edu.cn>.

*Proceedings of the 43rd International Conference on Machine Learning*, Seoul, South Korea. PMLR 306, 2026. Copyright 2026 by the author(s).

However, directly transplanting IM algorithms to SC is ill-suited because spatial tasks are inherently offline. Traditional IM and even Topic-Aware IM assume that propagation is driven primarily by semantic interests (Wang et al., 2025). For example, a user interested in technology is likely to engage with technology-related content online. However, spatial tasks require offline physical execution, imposing substantial spatial and physical costs that purely semantic topics cannot capture (Gao et al., 2025). In SC, influence does not strictly guarantee action because participation is constrained by the feasibility of the specific task instance. A community leader who holds significant sway in organizing local online gaming events may fail to mobilize their followers for a physical delivery task located miles away, despite the existence of strong social ties (Gao et al., 2024). This distinction implies that social influence in SC is not a universal scalar defined by abstract properties, but is tightly coupled to the heterogeneous attributes of specific tasks, such as location and reward. Therefore, we argue that effective recruitment must be Task-Aware, necessitating joint modeling of social bonds and the precise matching of workers to heterogeneous task instances, rather than relying on coarse-grained topical interests.

Furthermore, unlike viral marketing, in which maximizing the total number of influenced users is the sole objective, SC tasks impose strict quality requirements. Excessive recruitment for a single task leads to resource wastage and budget overruns, as the value of participation diminishes once the demand is met. Traditional utility maximization methods often overlook this saturation effect, simply accumulating potential contributions without regarding the demand cap (Deng, 2024). To address these challenges, we formulate the Effective Task Satisfaction (ETS) Maximization problem and prove that it is not submodular. This insight explains the failure of greedy algorithms and necessitates a new optimization approach. We propose GKD-Recruiter, a framework built on a Worker-Task Heterogeneous Graph. It employs Graph Knowledge Distillation to fuse directional social signals (via our novel Influential GAT) with physical task affinity, thereby creating robust, unified representations. We develop a Rainbow DQN algorithm to navigate the non-submodular search space. By leveraging long-term rewards, it avoids local optima associated with saturated tasks and achieves real-time inference by shifting computation to offline training. Extensive experiments on real-world datasets demonstrate that GKD-Recruiter significantly outperforms state-of-the-art baselines in solution quality while achieving orders-of-magnitude faster inference speed.

## 2. Related Work

**Social-Aware Worker Recruitment.** The problem of recruiting workers via social networks fundamentally stems from the classic Influence Maximization (IM) problem, which aims to maximize the number of activated nodes in a network. While foundational, traditional IM approaches focus solely on maximizing information spread, ignoring the specific quality and feasibility constraints of spatial tasks. To address semantic relevance, Topic-Aware IM methods, such as TIM-GNN, have been proposed to model user interests in specific topics (Halal et al., 2025). However, these methods remain ill-suited for Spatial Crowdsourcing due to the unique saturation trap inherent in spatial tasks. Both traditional and Topic-Aware IM algorithms typically operate under the assumption of submodularity, implying that recruiting more users always yields a marginal gain (Wang et al., 2024). In contrast, spatial tasks have finite demand capacities. Once a task is saturated, additional influence yields no marginal gain in effective task satisfaction. By ignoring this non-submodular constraint, coverage-centric methods often lead to inefficient budget allocation. Our work bridges this gap by explicitly optimizing for Effective Task Satisfaction, thereby handling both task heterogeneity and the demand saturation effect simultaneously.

**Reinforcement Learning in Spatial Crowdsourcing.** Deep Reinforcement Learning (DRL) has recently shown significant promise in solving complex combinatorial optimization problems within the spatial crowdsourcing domain (Ling et al., 2023). Notable recent contributions include DQNSelector, which utilizes DRL for worker recruitment, and DE-DQN, which addresses task assignment and route planning (Gao et al., 2024; 2025). While these works advance the state of the art, GKD-Recruiter differs fundamentally from them in terms of both granularity and methodology. Specifically, prior work, such as DQNSelector, typically selects seed workers at a coarse level (Gao et al., 2024). This approach overlooks the reality that a worker who is influential in one task type may be irrelevant in another. Our framework operates at the fine-grained worker-task pair level, enabling the precise matching of social influence to specific task instances. Furthermore, with respect to methodology, existing approaches often rely on shallow dual-embedding schemes to represent workers and tasks independently (Ma et al., 2024). In contrast, we propose a graph knowledge distillation framework that explicitly fuses heterogeneous signals from social and task perspectives, capturing higher-order interactions that simple concatenation strategies fail to model.

**Graph Learning for Heterogeneous Data.** Modeling spatial crowdsourcing data necessitates handling complex interactions among diverse entities, including users, tasks, and points of interest. While homogeneous Graph Neural Networks (GNNs) excel in modeling social influence, they struggle to capture the rich semantics of worker-task affinity. Heterogeneous Graph Neural Networks (HGNNs) offer a solution by modeling different node types and edge

relations (Jin et al., 2024). However, adapting HGNNs to the seed selection problem is non-trivial because it requires balancing social influence with task-matching requirements. Existing methods often treat these views independently or fuse them via simple linear layers, which may lead to information loss. To overcome this, our work introduces a teacher-student distillation paradigm. This approach enables the model to learn a robust, unified representation that preserves critical information from both the social graph and the worker-task heterogeneous graph (Tian et al., 2025), thereby ensuring more accurate and reliable seed selection.

# 3. Problem Formulation

In this section, we formulate the worker recruitment problem as a task-aware influence maximization task. Unlike traditional settings that select nodes to maximize generic influence, our goal is to select *worker-task pairs* to maximize the effective satisfaction of heterogeneous spatial tasks.

## 3.1. Task-Aware Influence Propagation

We consider a social network represented by a directed graph $\mathcal{G} = (\mathcal{V}, \mathcal{E}, \mathcal{W})$, where $v \in \mathcal{V}$ denotes a user, $e_{ij} \in \mathcal{E}$ is a social tie, and $w_{ij} \in \mathcal{W}$ represents the base influence probability from $v_i$ to $v_j$. The platform publishes a set of spatial tasks $\mathcal{T} = \{t_1, \ldots, t_M\}$. Each task $t_l \in \mathcal{T}$ is characterized by a spatial location $\ell_{t_l}$, a quality demand $d_l$, and a reward $r_l$.

**Task Affinity and Quality Potential.** Users exhibit different preferences and capabilities for distinct tasks. We define two key task-specific metrics for a user $v_i$ regarding task $t_l$:

- **Task Affinity ($a_i^l$):** Measures the willingness of user $v_i$ to propagate or participate in task $t_l$. It is derived from the user's historical interaction frequency and the task's reward $r_l$.

- **Quality Potential ($q_i^l$):** The potential quality increment user $v_i$ contributes to $t_l$. Following (Wang & Wu, 2021), we model this based on the Euclidean proximity between $v_i$'s trajectory history and task location $\ell_{t_l}$:

$$q_i^l = 1 - \frac{\min_r \|\ell_{i,r} - \ell_{t_l}\|_2}{\max_{v,k} \text{dist}(v, t_k)}, \tag{1}$$

where $\ell_{i,r}$ is $v_i$'s location at time step $r$, and the denominator is a normalization factor based on the maximum distance in the system.

**Task-Specific IC Model.** We extend the classic Independent Cascade (IC) model to a task-aware setting. We assume the propagation processes for different tasks are independent. Notably, the activation probability depends on both the social bond and the target user's affinity. Specifically,

an active user $v_i$ activates a neighbor $v_j$ for task $t_l$ with probability as follows:

$$p_{ij}(t_l) = w_{ij} \times a_j^l. \tag{2}$$

This formulation implies that influence only propagates effectively when the social connection is strong *and* the target user is interested in the specific task topic. Let $S \subseteq \mathcal{V} \times \mathcal{T}$ be the set of selected seed worker-task pairs. For a target user $v$, we denote $p_v(t_l, S)$ as the probability of being activated for task $t_l$ given seed set $S$.

## 3.2. Effective Task Satisfaction (ETS) Maximization

The core objective is to satisfy the quality requirements of tasks without resource waste (i.e., without oversatisfying demand $d_l$). For a given task $t_l$, the expected cumulative quality collected from all activated users is:

$$C(t_l, S) = \sum_{v \in \mathcal{V}} p_v(t_l, S) \cdot q_v^l. \tag{3}$$

To penalize redundant contributions beyond the demand $d_l$, we define the **Effective Task Satisfaction (ETS)** for task $t_l$ as the normalized valid contribution:

$$\text{ETS}(t_l, S) = \min\left(C(t_l, S)/d_l, 1\right). \tag{4}$$

**Problem Definition (ETS-Maximization).** Given the social graph $\mathcal{G}$, task set $\mathcal{T}$, and a budget $K$, we aim to select a seed set $S$ of at most $K$ worker-task pairs to maximize the global effective task satisfaction:

$$\max_{S \subseteq \mathcal{V} \times \mathcal{T}} \frac{1}{M} \sum_{l=1}^{M} \text{ETS}(t_l, S) \tag{5}$$

$$\text{s.t.} \quad |S| \leq K, \text{count}(v, S) \leq U_{max}, \forall v \in \mathcal{V}$$

where $\text{count}(v, S)$ denotes the number of tasks assigned to user $v$, constrained by a maximum workload $U_{max}$.

## 3.3. Theoretical Analysis

**Theorem 3.1 (NP-Hardness).** *The ETS-Maximization problem defined in Eq. 5 is NP-hard.*

*Proof.* The detailed proof is provided in Appendix E. □

**Theorem 3.2 (Objective Intractability).** *Computing the exact value of the objective function $\mathcal{J}(S) = \mathbb{E}[\min(C(S)/d, 1)]$ is #P-hard.*

*Proof.* Computing the exact expected spread in the standard IC model is #P-hard (Chen et al., 2010). Our objective is even more complex due to the truncation function $\min(\cdot, 1)$. Calculating $\mathbb{E}[\min(X, d)]$ requires knowledge of the specific distribution of the random variable $X$ (the cumulative quality), not just its expectation $\mathbb{E}[X]$. Since computing the

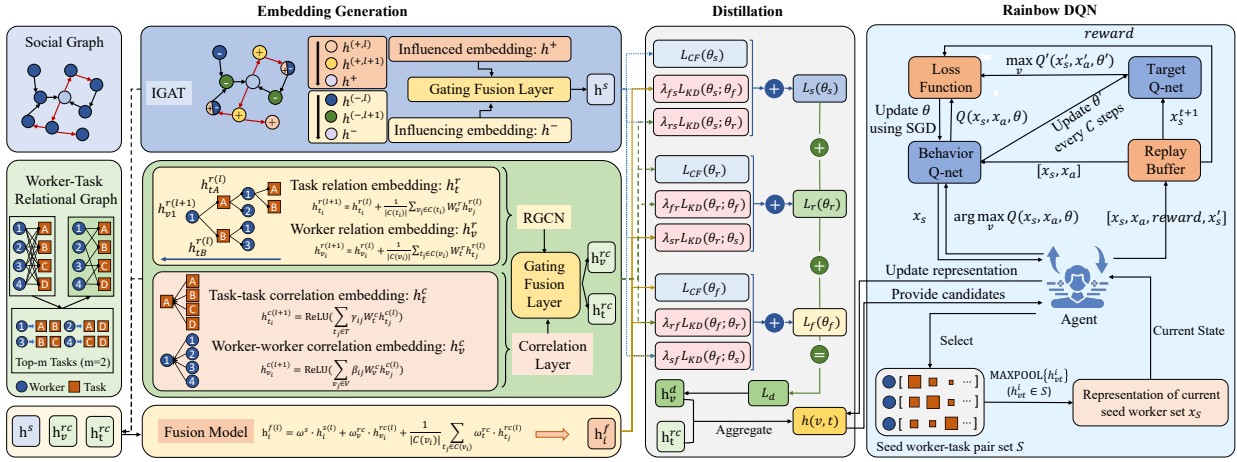

*Figure 2.* The architecture of GKD-Recruiter. (1) **Social View**: IGAT captures bi-directional influence flow. (2) **Task View**: RGCN and Correlation Layers model worker-task affinity and intra-type similarities. (3) **Distillation**: A fusion module integrates views via Knowledge Distillation. (4) **Decision**: Rainbow DQN selects seed pairs iteratively based on the learned state representations.

probability distribution of the number of activated nodes in a general graph is #P-hard, computing our exact objective is also #P-hard. This intractability necessitates the use of Monte Carlo (MC) simulations or learning-based approximations (like our proposed RL approach) rather than exact analytical solutions. □

**Proposition 3.3** (**Suboptimality of Greedy Strategy under Capacity Constraints**). *Under the finite workload capacity constraint* ($count(v, S) \leq U_{max}$)*, the greedy strategy fails to guarantee an optimal solution. It ignores the opportunity cost of allocating a versatile worker's limited capacity to tasks that could be performed by less capable workers.*

*Proof.* We rigorously prove this in Appendix F by constructing a counterexample where the greedy strategy achieves only 50% of the optimal utility. □

**Algorithmic Implications and Motivation.** The theoretical suboptimality in Proposition 3.3 exposes a critical flaw in traditional heuristics like CELF. These algorithms rely on the submodularity assumption, expecting locally optimal choices to converge to a global optimum. However, in Spatial Crowdsourcing, the interplay between *Task Heterogeneity* and finite *Capacity Constraints* creates a "Resource Contention Trap." Greedy methods ignore opportunity costs, prematurely locking versatile workers into suboptimal tasks, thereby creating bottlenecks for harder tasks.

Furthermore, computational efficiency poses a prohibitive bottleneck. Simulation-based methods (e.g., CELF, TIM) rely on extensive Monte Carlo sampling to estimate marginal gains, resulting in a time complexity of $\mathcal{O}(K \cdot |V| \cdot R)$. For modern platforms requiring millisecond-level responsiveness, such latency is unacceptable.

These dual challenges necessitate a shift to Deep Reinforcement Learning (DRL). DRL addresses these issues by: (1) optimizing long-term cumulative rewards to identify when to "reserve" versatile workers, thus avoiding saturation traps; and (2) shifting the heavy computational burden to the offline training phase to enable efficient real-time inference. This motivates our GKD-Recruiter framework.

## 4. GKD-Recruiter Design

In this section, we present **GKD-Recruiter**, a framework that integrates heterogeneous graph learning with reinforcement learning for seed selection. As illustrated in Figure 2, the model operates in two phases: *representation learning* via graph knowledge distillation, and *seed selection* via Rainbow DQN. To handle raw attributes, we first employ MLPs to project worker and task features into $d$-dimensional embeddings, which serve as inputs to subsequent modules.

### 4.1. Bi-directional Social Influence Modeling (IGAT)

Social influence is inherently directional and asymmetric. Unlike standard GNNs that aggregate neighbors indiscriminately, we propose IGAT to capture the distinct roles of influencing ($N_i^{out}$) and being influenced ($N_i^{in}$). To align the representation learning with the physical diffusion process, we condition the attention mechanism on the base propagation probability $w_{ij}$. The attention coefficient $e_{ij}^{dir}$ for a direction $dir \in \{in, out\}$ is computed as:

$$e_{ij}^{dir} = \text{LeakyReLU}\left(\mathbf{a}_{dir}^{T}[\mathbf{W}h_i^{(l)} \,\|\, \mathbf{W}h_j^{(l)} \,\|\, w_{ij}]\right), \quad (6)$$

where $\|$ denotes concatenation. This formulation allows the model to prioritize neighbors who are not only semantically similar but also structurally influential in the cascade. Once normalized coefficients $\alpha_{ij}^{dir}$ are obtained via the softmax

function, neighbor aggregation for a specific direction $dir \in \{in, out\}$ is formally defined as:

$$h_i^{dir} = \sigma \left( \sum_{j \in \mathcal{N}_i^{dir}} \alpha_{ij}^{dir} \mathbf{W} h_j^{(l)} \right), \qquad (7)$$

where $\sigma$ denotes a non-linear activation function (e.g., ELU), $\mathcal{N}_i^{dir}$ represents the directional neighborhood of worker $v_i$, and $\mathbf{W}$ is the shared learnable weight matrix. The resulting view-specific embeddings $h_i^{in}$ and $h_i^{out}$ are then passed into the gating fusion layer. Finally, we fuse the bi-directional embeddings using a learnable gating mechanism to preserve the most informative signals from both views:

$$h_i^s = [G^{in} \odot h_i^{in} \,||\, G^{out} \odot h_i^{out}]. \qquad (8)$$

## 4.2. Heterogeneous Worker-Task Modeling

To capture the compatibility between workers and tasks, we construct a pruned worker-task heterogeneous graph $\mathcal{G}_r$. To reduce computational complexity, we link each worker only to their top-$m$ tasks with the highest potential quality (Eq. 1). We employ a Relational GCN (RGCN) to aggregate heterogeneous neighbors. Let $\mathcal{N}_i^t$ be the connected tasks for worker $v_i$. The RGCN update is:

$$h_{v_i}^{r(l+1)} = h_{v_i}^{r(l)} + \frac{1}{|\mathcal{N}_i^t|} \sum_{t_j \in \mathcal{N}_i^t} \mathbf{W}_r h_{t_j}^{r(l)}. \qquad (9)$$

A symmetric update is applied for task nodes $h_{t_i}^r$.

**Correlation Layer.** Beyond direct worker-task edges, capturing global correlations (e.g., workers with similar skill sets or tasks with similar patterns) is crucial. We introduce a correlation layer:

$$h_{v_i}^{c(l+1)} = \text{ReLU} \left( \sum_{v_j \in V} \beta_{ij} \mathbf{W}_c h_{v_j}^{c(l)} \right), \qquad (10)$$

where $\beta_{ij}$ denotes the normalized similarity score between worker $v_i$ and $v_j$. Task embeddings $h_{t_i}^c$ are updated similarly. Finally, we fuse the RGCN embedding ($h^r$) and Correlation embedding ($h^c$) via a gating layer to obtain the final task-view representations $h_{v_i}^{rc}$ and $h_{t_i}^{rc}$.

## 4.3. Graph Knowledge Distillation

We derive two view-specific embeddings for each worker: $h_i^s$ from the social view and $h_i^{rc}$ from the worker-task view. Merely concatenating these vectors forces the model to treat all signals equally, which risks allowing noise from one view to dominate the final representation. To robustly fuse these heterogeneous signals, we employ a Knowledge Distillation framework (Wu et al., 2019) that learns a unified representation through mutual supervision.

We first construct a Fusion Model to serve as the teacher. Instead of relying on static weights, we employ a standard

gating mechanism to dynamically balance the signals based on the specific context of the worker. The fused representation $h_i^f$ is computed as:

$$h_i^f = G_s \odot h_i^s + G_{rc} \odot h_i^{rc} + G_n \odot \text{Mean}(h_{\mathcal{N}_i}^{rc}), \quad (11)$$

where $G_k$ represents learnable element-wise gates derived from the concatenated features. This mechanism allows the model to adaptively highlight informative dimensions, effectively distinguishing between social influence and task affinity based on the input.

Notably, the value of this framework extends beyond simple feature aggregation. The fusion model acts as a teacher to guide the single-view student models via mutual learning. This process serves as a form of regularization, forcing the final representation to retain consensus information across views while suppressing view-specific noise. The joint objective function is defined as:

$$\mathcal{L} = \sum_{m \in \{f,s,r\}} \mathcal{L}_{CF}(\theta_m) + \lambda \sum_{m \in \{f,s,r\}} \sum_{k \neq m} \mathcal{L}_{KD}(\theta_m; \theta_k),$$

$$(12)$$

where $\mathcal{L}_{CF}$ is the prediction loss, such as BPR, and $\mathcal{L}_{KD}$ minimizes the discrepancy between student and teacher embeddings. Consequently, the final distilled embedding $h_i^d$ derived from the fusion model is structurally robust and semantically rich.

## 4.4. Seed Selection via Rainbow DQN

We formulate the seed selection problem as a Markov Decision Process (MDP) and employ Rainbow DQN (Hessel et al., 2018) to optimize the policy. The core components are defined as follows:

**State ($s_t$):** Represents the set of currently selected seed worker-task pairs $S^t$. We aggregate the individual pair embeddings using a max-pooling operation to obtain the fixed-length state vector: $x_s^t = \text{MAXPOOL}\{h_{vt}^i \mid h_{vt}^i \in S^t\}$, where $h_{vt}$ denotes the pair representation.

**Action ($a_t$):** Corresponds to selecting a candidate worker-task pair $\langle v, t \rangle$. To obtain its representation $h_{(v,t)}$, we fuse the distilled worker embedding $h_v^d$ and task embedding $h_t^{rc}$ using the same gating layer structure as the above one.

**Reward ($r_t$):** Defined as the marginal gain in the overall effective task satisfaction (ETS) achieved by adding the new pair. Formally, $r_t = \text{ETS}(S^t \cup \{a_t\}) - \text{ETS}(S^t)$.

**Policy:** We follow the standard training protocol of Rainbow DQN. The complete training algorithm is detailed in Algorithm 1 (Appendix G).

# 5. Experiments

## 5.1. Datasets and Experiment Setups

We conduct experiments on two real-world LBSN datasets: **Gowalla** (12,741 users, 101,394 edges, 620k check-ins) and **Brightkite** (8,573 users, 61,574 edges, 516k check-ins). Following (He et al., 2015), we discretize time into 2-hour cycles. For the social network, we sample subgraphs with $|V| \in \{3000, 5000\}$ preserving power-law degree distributions. The worker pool size is $|U| = 300$, and the task set $|T| = 100$. The budget $|S|$ varies in $\{25, 50, \dots, 150\}$. Task demand $d_l$ is determined by local trajectory density. The Quality Potential $q_i^l$ and Task Affinity $a_i^l$ are modeled based on Euclidean proximity and historical visitation frequency, respectively:

$$q_i^l = 1 - dis_i(t_l)/D_{max}, a_i^l = r_l \cdot n(v_i, t_l)/N_{max}, \quad (13)$$

where $D_{max}$ and $N_{max}$ are normalization factors. We train GKD-Recruiter for 500 episodes with $|S| = 75$, using a 9:1 train-test split. The default candidate size is $m = 5$.

## 5.2. Baseline Algorithms

To strictly evaluate the performance of GKD-Recruiter, we compare it against seven representative baselines, categorized into three groups:

**1. Heuristic Approaches.** These methods select seeds based on predefined centrality or heuristic rules.

- **DegGreedy**: Selects the top-$K$ worker-task pairs solely based on the nodal degree, assuming high-degree nodes have better propagation capabilities.
- **NDD (Node Degree Decay)** (Chen et al., 2009): Mitigates influence overlap by dynamically attenuating the out-degree of a node based on the current seed set intersections.
- **FastSelector** (Wang et al., 2019): A domain-specific heuristic that ranks users via a hybrid score $R(v) = \alpha \cdot DR(v, S) + (1 - \alpha) \cdot TR(v, S)$, combining degree rank and trajectory difference.
- **ComGreedy**: A metric-based greedy method that selects pairs with the highest single-step effective task satisfaction ($com$). The score for node $v_i$ is: $com(v_i) = \frac{1}{|T|} \sum_{t_l \in T} \sum_{v_j \in N_+(v_i)} w_{ij} \cdot a_j^l \cdot q_j^l$.

**2. Optimization-based Approaches (IM).** These adapt classic Influence Maximization algorithms.

- **CELF** (Leskovec et al., 2007): Accelerates greedy selection using a "lazy forward" strategy based on submodularity. *Note: Its theoretical guarantee holds for submodular objectives but fails for our non-submodular ETS objective.*
- **TSIM** (Qiu et al., 2020): A two-stage framework combining degree decay with delayed forward selection and neighbor pruning to accelerate MC simulations.

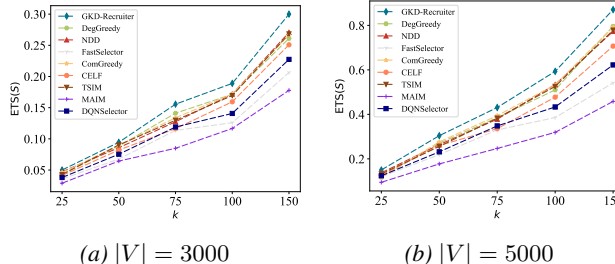

*(a) $|V| = 3000$*       *(b) $|V| = 5000$*

*Figure 3.* Effective Task Satisfaction (ETS) vs. Number of Seeds ($K$). GKD-Recruiter consistently achieves higher ETS across varying budgets.

**3. RL-based Approaches.**

- **MAIM** (Liu et al., 2021): A multi-agent RL approach using DQN with memory isolation. It relies on homogeneous graph structures and ignores heterogeneous worker-task affinities.
- **DQNSelector** (Gao et al., 2024): A state-of-the-art DRL-based worker recruitment framework. It utilizes a dual-embedding scheme to represent workers and tasks separately and employs a DQN agent for seed selection.

## 5.3. Performance Analysis and Comparison

**Effectiveness Analysis.** Figure 3 illustrates the Effective Task Satisfaction (ETS) performance as the seed budget $K$ increases on both datasets. The results demonstrate that GKD-Recruiter consistently outperforms all baselines by a significant margin. Specifically, on the larger dataset ($|V| = 5000$), our method surpasses the strongest heuristic method, ComGreedy, by 11.46%, and the classic Influence Maximization algorithm, CELF, by 21.41%. It is worth noting that in large-scale spatial crowdsourcing platforms, an improvement of this magnitude implies a substantial increase in economic utility and task fulfillment.

The superiority of GKD-Recruiter can be attributed to its ability to overcome the limitations inherent in traditional approaches. A critical observation is the relatively poor performance of CELF and other greedy-based methods. These algorithms are designed under the assumption of submodularity, expecting that the marginal gain of adding a seed always diminishes gradually. However, as discussed in Section 3.3, the ETS objective is non-submodular due to the demand saturation constraint. Greedy algorithms fail to anticipate this "saturation trap" and often over-commit to tasks that are easily saturated, resulting in budget wastage. In contrast, GKD-Recruiter employs a reinforcement learning agent trained with long-term cumulative rewards, enabling it to look ahead and diversify selections to avoid saturation.

Furthermore, compared with MAIM and DQNSelector, RL-based methods, GKD-Recruiter achieves a significant

improvement. This highlights the advantage of our fine-grained modeling. MAIM relies on coarse-grained worker embeddings and ignores the specific affinity between a worker and a task. DQNSelector typically treats worker and task representations as separate entities combined via simple interactions. Our framework explicitly models pair-wise interactions via the RGCN and distillation modules, ensuring that the selected seeds are not only socially influential but also strictly compatible with the spatial tasks.

Finally, we observe that the performance gap between GKD-Recruiter and the baselines widens as the number of seeds increases. This indicates that while heuristic methods can identify "obvious" high-quality seeds (low-hanging fruits), they struggle as the budget grows. GKD-Recruiter, however, successfully uncovers "non-obvious" yet high-potential worker-task pairs, demonstrating robust generalization capabilities even when the inference budget differs from the training setting ($K = 75$).

### 5.4. Efficiency and Scalability Analysis

Beyond solution quality, the computational efficiency of the inference phase is critical for real-time spatial crowdsourcing applications. We evaluate the scalability of GKD-Recruiter against baselines, with a specific focus on the larger dataset ($|V| = 5000$) to highlight performance under stress. Table 1 details the running time required for seed selection across varying budgets.

*Table 1.* Inference Time (sec.) on Large Dataset ($|V| = 5K$)

| Model \ $|S|$ | 25 | 50 | 75 | 100 | 150 |
|---|---|---|---|---|---|
| GKD-Recruiter | 0.97 | 1.94 | 3.30 | 3.92 | 6.34 |
| DegGreedy | 0.08 | 0.11 | 0.23 | 0.45 | 0.72 |
| NDD | 0.46 | 0.88 | 1.40 | 1.16 | 2.65 |
| FastSelector | 2.47 | 5.39 | 15.47 | 20.39 | 26.28 |
| ComGreedy | 16.52 | 35.97 | 54.47 | 69.78 | 255.22 |
| CELF | 12.76 | 23.73 | 38.18 | 32.56 | 72.14 |
| TSIM | 2.58 | 4.62 | 7.58 | 6.49 | 14.15 |
| MAIM | 1.59 | 3.29 | 5.09 | 6.58 | 10.47 |
| DQNSelector | 1.62 | 3.35 | 5.18 | 6.72 | 10.85 |

The empirical results demonstrate that GKD-Recruiter achieves orders-of-magnitude acceleration compared to simulation-based and metric-based heuristics. As the seed budget increases to 150, methods like ComGreedy and CELF exhibit exponential growth in runtime, reaching 255.22s and 72.14s respectively, rendering them impractical for online deployment. In contrast, GKD-Recruiter completes the selection process in just 6.34 seconds, maintaining real-time responsiveness. Even compared to MAIM, another RL-based approach, our method is approximately 1.6 times faster while delivering significantly higher solution quality (as established in Section 5.2).

This efficiency advantage is theoretically grounded in the algorithmic complexity. For greedy-based baselines like CELF, the time complexity is dominated by Monte Carlo simulations required to estimate the objective function, scaling as $\mathcal{O}(K \cdot |V| \cdot R)$, where $R$ (typically $10^4$) is the number of simulations. This heavy sampling burden creates a computational bottleneck. Conversely, GKD-Recruiter shifts the heavy lifting to the offline training phase. During online inference, the complexity is reduced to $\mathcal{O}(K \cdot (|V| + |E|))$, which primarily involves efficient matrix operations on the GPU. Since $R \gg |E|/|V|$ in typical social graphs, GKD-Recruiter is inherently more scalable. The results on the smaller dataset ($|V| = 3000$) follow a consistent trend and are provided in Appendix H.

### 5.5. Ablation Study

To assess the contribution of each component in GKD-Recruiter, we conducted a comprehensive ablation study. We introduced four variants: (1) **w/o IGAT**, replacing the Influential GAT with a standard GAT; (2) **w/o RGCN**, removing the Relational GCN module; (3) **w/o Corr.**, removing the correlation layer; and (4) **w/o Dist.**, removing the distillation loss. Table 2 summarizes the results with a budget of 100 seeds.

*Table 2.* Ablation Analysis: ETS with 100 Seeds

| Dataset | w/o IGAT | w/o RGCN | w/o Corr. | w/o Dist. | **GKD-Rec.** |
|---|---|---|---|---|---|
| $|V| = 3K$ | 0.2314 | 0.2456 | 0.2694 | 0.2714 | **0.2833** |
| $|V| = 5K$ | 0.4704 | 0.4907 | 0.5537 | 0.5561 | **0.5933** |

**Impact of Heterogeneous Worker-Task Modeling (RGCN).** As observed in Table 2, removing the RGCN module results in the most significant performance degradation (e.g., a 17.3% drop on the $|V| = 5000$ dataset). This indicates that RGCN is the backbone of our framework. Its primary role is to encode the *Task Affinity* by distinguishing between worker-task edges and social edges in terms of semantic differences. We also attempted to replace RGCN with a standard homogeneous GCN. However, the model failed to converge. This failure highlights that treating diverse relationships (social ties vs. task matching) uniformly introduces severe noise, preventing the RL agent from learning a stable policy.

**Impact of Bi-directional Social Influence (IGAT).** Replacing IGAT with a standard GAT or GraphSAGE leads to a noticeable performance drop (from 0.2833 to 0.2314 on $|V| = 3000$). Standard GNNs aggregate neighbor information indiscriminately. In contrast, our IGAT explicitly models the asymmetry of social power by distinguishing between "influencing" (out-neighbors) and "being influenced" (in-neighbors) roles. This structural distinction is critical for identifying seed users who can effectively trigger information cascades.

**Impact of Fusion Mechanisms (Correlation & Distillation).** The removal of the Correlation Layer or the Distillation Module leads to a performance decrease of approximately 4-6%. While each view (Social or Task) provides partial information, the distillation module acts as a regularization mechanism. It forces the fused representation to retain the most informative signals from both views while suppressing view-specific noise. This creates a more robust state representation for the Rainbow DQN, enabling it to generalize better across different network scales.

## 5.6. Robustness to Dynamic Environments

Real-world social networks are highly dynamic, with edge structures and influence probabilities evolving over time. To verify the robustness of GKD-Recruiter, we simulate a dynamic scenario on the $|V| = 5000$ dataset. Specifically, we randomly perturb 5% of the edges in the social network, modifying both linkage and influence weights.

For the baselines such as ComGreedy and CELF, adapting to these changes requires full re-execution from scratch, which is computationally expensive. In contrast, for GKD-Recruiter, we employ a Rapid Fine-tuning strategy: we freeze the main RL policy and update only the GNN parameters associated with the changed local structures.

*Table 3.* Robustness under Network Perturbation ($K = 50$)

| Method | Original ETS | Perturbed ETS | Adaptation Cost |
|---|---|---|---|
| **GKD-Rec.** | **0.3042** | **0.2938** | **Low (Fine-tuning)** |
| DegGreedy | 0.2698 | 0.2767 | Low (Heuristic) |
| NDD | 0.2636 | 0.2718 | Low (Heuristic) |
| FastSelector | 0.2189 | 0.2213 | High |
| ComGreedy | 0.2762 | 0.2895 | High (Re-calculation) |
| CELF | 0.2598 | 0.2694 | Very High (Re-simulation) |
| TSIM | 0.2559 | 0.2710 | High |
| MAIM | 0.1773 | 0.1845 | High (Retraining) |
| DQNSelector | 0.2315 | 0.2250 | Low (Fine-tuning) |

The results in Table 3 reveal a critical insight. While baselines achieve slightly improved performance, they do so at the cost of expensive full re-computation. GKD-Recruiter experiences a marginal performance drop ($\sim$3.4%) because it does not relearn the entire graph from scratch. Notably, even after this slight drop, GKD-Recruiter (0.2938) still outperforms the best fully retrained baseline (ComGreedy at 0.2895). This demonstrates that GKD-Recruiter achieves an optimal trade-off: it maintains state-of-the-art performance while adapting to dynamic environments with minimal computational overhead.

## 5.7. Parameter Sensitivity

We investigate the sensitivity of GKD-Recruiter to two key hyperparameters: the number of GNN layers ($L$) and the size of the candidate task set ($m$).

**Impact of GNN Layers ($L$).** Table 4 reports the perfor-

*Table 4.* Sensitivity of GNN Layers ($|V| = 3K, K = 100$)

| Layers ($L$) | 1 | 2 | 3 |
|---|---|---|---|
| ETS | **0.2833** | 0.2776 | 0.2801 |

*Table 5.* Sensitivity of Candidate Size $m$ ($|V| = 3K, K = 100$)

| Top-$m$ Tasks | $m = 3$ | $m = 5$ | $m = 7$ | $m = 9$ |
|---|---|---|---|---|
| ETS | 0.2712 | **0.2833** | 0.2815 | 0.2798 |
| Time Cost | Low | Medium | High | Very High |

mance as we stack more IGAT and RGCN layers. Theoretically, deeper GNNs can capture longer-range dependencies. However, in our dense social graph scenario, we observe that $L = 1$ achieves the best performance (0.2833). Increasing $L$ to 2 or 3 leads to performance degradation. This is likely due to the over-smoothing problem common in GNNs, in which node representations become indistinguishable as depth increases, thereby introducing noise into the RL state representation. Thus, we set $L = 1$ for efficiency and effectiveness.

**Impact of Candidate Task Size ($m$).** The parameter $m$ determines the sparsity of the worker-task heterogeneous graph (connecting each worker to their top-$m$ potential tasks). We conducted a grid search for $m \in \{3, 5, 7, 9\}$, as shown in Table 5.

The results exhibit an inverted U-shape trend.

- **Under-fitting ($m = 3$):** The graph is too sparse, causing the model to miss potential high-quality worker-task matches, leading to lower ETS (0.2712).
- **Complexity Overload ($m > 5$):** As $m$ increases, the action space for the RL agent expands significantly. This not only increases training time but also introduces irrelevant task edges (noise), making it harder for the agent to converge to an optimal policy.

Therefore, $m = 5$ strikes the optimal balance between preserving critical information and maintaining efficiency.

## 6. Conclusion

In this paper, we address the challenge of Task-Aware worker recruitment in spatial crowdsourcing. We formulate the problem as maximizing Effective Task Satisfaction, distinguishing our approach from traditional influence maximization by explicitly modeling the heterogeneity of physical tasks. To solve this, we propose GKD-Recruiter, a novel framework that integrates heterogeneous graph learning, knowledge distillation, and Rainbow DQN to identify optimal seed worker-task pairs. Extensive experiments on real-world datasets demonstrate that GKD-Recruiter significantly outperforms state-of-the-art baselines.

## Acknowledgments

This work was partially supported by the National Key Research and Development Program of China [2024YFF0617700]; the National Natural Science Foundation of China [U23A20309, 62272302, U22A2025, 62232007, and U23A20309]; the 111 Project [B16009]; the Guangdong Basic and Applied Basic Research Foundation [2025A1515012843]; the Ant Group Research Program [2025021900003]; the Fundamental Research Funds for the Central Universities [N2417007]. We also thank Jiale Zhang for the contribution to collecting information on crowdsourcing platforms and ICML-related papers in the appendix.

## Impact Statement

This paper presents work whose goal is to advance the field of machine learning. There are many potential societal consequences of our work, none of which we feel must be specifically highlighted here.

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

## A. Summary of Social Recruitment in Crowdsourcing Platforms

Table 6 summarizes real-world spatial crowdsourcing platforms that integrate social mechanisms for worker recruitment, as discussed in Section 1.

*Table 6.* Summary of Spatial Crowdsourcing Platforms and Task Types

| Category | Platform Name | Task Type |
|---|---|---|
| **Crowd Sensing** | CrowdOS | Integrated mobile sensing: environmental monitoring, traffic flow collection, and location check-in. |
| **Environmental** | Noise Tube | Urban noise data collection and mapping. |
| | Common Sense | Air quality and environmental parameter monitoring. |
| | Ear-Phone | Noise pollution sensing and distribution analysis. |
| | Creek Watch | Photo reporting of river and water quality. |
| **Transport** | CrowdWatch | Bus operation status and road congestion detection. |
| | CrowdNavi | Real-time road condition reporting and navigation. |
| | Mapillary | Street view capture via smartphone cameras with automated geographic info extraction. |
| | PhotoCity | Urban street view photo collection and annotation. |
| | OpenStreetMap | Crowdsourced map layer editing and POI annotation. |
| | Strava Metro | Aggregation of user GPS trajectories (sports/commuting) for urban transport planning. |
| | StreetComplete | Filling missing data items in open-source maps (street facilities, road attributes, etc.). |
| **Public Safety** | IFRC GO | Disaster response and information management (monitoring and info sharing). |
| | Earthquake Network | Early earthquake detection and warning using smartphone accelerometer data. |
| | Ushahidi | Geographic event reporting: Crisis/disaster site information collection and visualization. |
| | Crowd Tracking | Personnel localization and crowd behavior analysis. |
| **Smart Mobility** | Uber | Ride-hailing and trip scheduling. |
| | Didi Chuxing | Ride-hailing, carpooling, and designated driving services. |
| **Place Review** | Foursquare | Venue check-ins, reviews, and information editing. |
| | Dianping | Reviews and ratings for dining, entertainment, and lifestyle services. |
| | Mafengwo | Travel route check-ins and review collection. |
| | TripAdvisor | Tourist attraction check-ins and review collection. |
| **Food Delivery** | Taobao Flash | Food ordering and delivery scheduling. |
| | Meituan Delivery | Diversified catering and fresh food delivery services. |
| **Game** | Pokémon GO | Location-based data collection through gameplay. |
| | Ingress | Real-world location capturing and territorial control via GPS. |
| | Zombies, Run! | Fitness tracking during outdoor exercise. |

## B. Visualization of Social Propagation Modes

To further illustrate the social recruitment mechanisms discussed in Section 1, Figure 4 visualizes the three primary channels through which task information propagates in modern Spatial Crowdsourcing (SC) platforms.

As depicted in Figure 4, the social connections in SC are not limited to a single graph but manifest in hybrid forms:

- **Internal Interaction (Left):** Users communicate directly within the crowdsourcing app via built-in chat rooms or

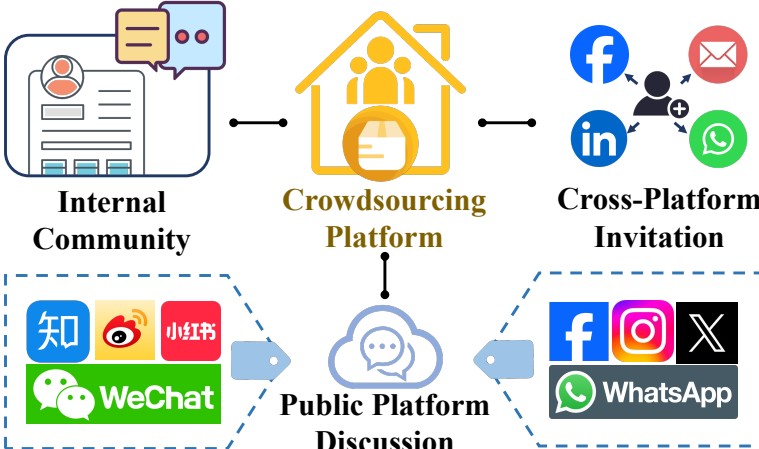

*Figure 4.* Schematic illustration of multi-channel social propagation patterns in Spatial Crowdsourcing. Information diffuses through (1) Internal Interactions within the platform, (2) Broad discussions on Public Communities (e.g., Reddit, XiaoHongShu), and (3) Targeted Cross-Platform Invitations via social media (e.g., WhatsApp, WeChat).

forums to coordinate on co-located tasks (e.g., team-based gaming in Pokémon GO).

- **Public Community Discussion (Center):** Task information spreads through topic-based communities on public platforms. For instance, workers may discuss high-reward tasks on specialized forums or lifestyle apps (e.g., XiaoHongShu or Reddit), forming implicit interest groups.

- **Cross-Platform Invitation (Right):** This is the most direct form of recruitment, where existing workers utilize external social networks (e.g., WhatsApp , WeChat, LinkedIn) to send targeted invitations to their acquaintances, often incentivized by referral bonuses (e.g., Uber, Meituan).

This multi-channel propagation structure justifies our design of the Worker-Task Heterogeneous Graph, which aims to capture these diverse and complex interaction signals beyond simple topological connections.

## C. Influence Maximization

For a given social network, we can represent it as a directed graph $G = (V, E)$, where the set of nodes $V = \{v_1, v_2, \cdots, v_{|V|}\}$ and the set of edges $E = \{e_1, e_2, \cdots, e_{|E|}\}$ are defined. Each node $v$ in $V$ signifies a user, and each edge $e = (u, v)$ in $E$ symbolizes a certain type of connection from user $u$ to $v$. For each edge $(u, v) \in E$, $u$ is termed as the in-neighbor of $v$, and conversely, $v$ is the out-neighbor of $u$. For any given node $v$ in $V$, the collection of its in-neighbors is denoted as $N^-(v)$, and the collection of its out-neighbors is denoted as $N^+(v)$.

In the process of information diffusion, a user is deemed active when she accepts (is activated by) the information from her in-neighbors or is chosen as an initial influencer (seed). This information cascade is often characterized by a diffusion model, for instance, the Independent Cascade (IC) model (Kempe et al., 2003).

**Definition C.1** (IC model). Under the IC model, with an initial seed set $S \subseteq V$, the process of stochastic information spread is as follows: (1) Initially, at timestamp 0, all nodes in $S$ are in an active state, while the rest of the nodes in $V \backslash S$ remain inactive. Once activated, a node stays active; (2) At any timestamp $\tau$, every newly activated node $u$ gets a single opportunity to switch its inactive out-neighbor $v$ to an active state with a probability of $p_{uv}$ at timestamp $\tau + 1$; and (3) The process of information spreading concludes when there are no further inactive nodes that can be activated in the upcoming timestamps.

Subsequently, we take the IC model $\mathcal{D}(G, \boldsymbol{p})$ as an example to discuss the IM problem, which can be parameterized by a parameter set $\boldsymbol{p} = \{p_{uv} : (u, v) \in E\}$ given a social graph $G$. (When the context is clear, the IC model can be abbreviated as $\mathcal{D}$.) A realization, denoted as $g = (V, E_g)$ where $E_g \subseteq E$, represents a subgraph that is randomly generated based on the diffusion model. Taking the IC model as an instance, each edge $(u, v) \in E$ is independently included in $E_g$ with a likelihood of $p_{uv}$. Edges contained in $E_g$ in the realization $g$ are termed "live" edges. Thus, the probability of realization

$g$ sampled based on the IC model $\mathcal{D}(G, \boldsymbol{p})$ is $\Pr[g; \mathcal{D}(G, \boldsymbol{p})] = \prod_{e \in E_g} p_e \prod_{e \in E \setminus E_g} (1 - p_e)$. Considering there are $2^{|E|}$ potential realizations, the influence cascade in any realization is deterministic, not stochastic. Therefore, the influence spread through the network is essentially the average of the spread across all possible realizations.

**Definition C.2** (Influence Maximization). In the context of a social graph $G = (V, E)$, considering a diffusion model (specifically, the IC model discussed in this chapter), and given a limit of $K$ nodes (referred to as the budget), the IM problem is to identify an optimal seed set $S^\circ$ with at most $K$ nodes that is intended to maximize the expected influence spread throughout the network, i.e.

$$S^\circ \in \arg\max_{|S| \leq K} \sigma_{\mathcal{D}}(S) = \mathbb{E}_{g \sim \mathcal{D}(G, \boldsymbol{p})}[|I_g(S)|] = \sum_{g \in \mathcal{G}} \Pr[g; \mathcal{D}(G, \boldsymbol{p})] \cdot |I_g(S)|, \tag{14}$$

where $\mathcal{G}$ is the ensemble of all conceivable realizations and $I_g(S)$ is the set of nodes that encompasses all nodes that are reachable from any node in $S$ through the live edges in the realization $g$.

## D. Notation Table

*Table 7.* Symbols and Definitions

| Symbol | Definition |
|---|---|
| $G = (V, E, W)$ | directed social graph |
| $v_i \in V$ | social user |
| $e_{ij} \in E$ | social relationship from $v_i$ to $v_j$ |
| $w_{ij} \in W$ | social influence of $v_i$ to $v_j$ |
| $\mathcal{T} = \{t_1, \ldots, t_M\}$ | crowdsourcing task set, $t_l$ represents $l_{th}$ task |
| $d_l$ | completion quality demand for task $t_l$ |
| $c_l$ | cumulative completion of task $t_l$ |
| $a_i = \{a_i^1, \cdots, a_i^{|T|}\}$ | task affinity of different tasks to $v_i$ |
| $q_i = \{q_i^1, \cdots, q_i^{|T|}\}$ | completion quality of $v_i$ on different tasks |
| $U$ | worker pool |
| $S$ | seed worker-task pair set |
| $K$ | quantity limit of seed pair selection |
| $N_i^-, N_i^+$ | in-neighbors and out-neighbors of $v_i$ |
| $p_i(t_l, S)$ | influenced probability of $v$ on $t_l$ under $S$ |
| $q_i(t_l, S)$ | completion quality of $v_i$ on $t_l$ under $S$ |
| $C(t_l, S)$ | cumulative task satisfaction for $t_l$ under $S$ |
| $\text{ETS}(t_l, S)$ | effective task satisfaction for $t_l$ under $S$ |
| $\text{ETS}(S)$ | overall effective task satisfaction |

## E. Proof of Theorem 3.1

*Proof.* We prove this by reduction from the classic Influence Maximization (IM) problem, which is known to be NP-hard (Kempe et al., 2003). Consider a special instance of our problem where: (1) The task set contains only one task $t_1$; (2) The demand $d_1$ is set to infinity ($d_1 \to \infty$); (3) The quality potential $q_v^1 = 1$ for all users. In this setting, the Effective Task Satisfaction term $\min(C(S)/d, 1)$ simplifies to the cumulative probability sum $C(S)$, which is equivalent to the expected spread in the IM problem. Since any instance of the NP-hard IM problem can be reduced to this instance of ETS-Maximization, our problem is at least as hard as IM. Thus, ETS-Maximization is NP-hard. $\square$

## F. Proof of Proposition 3.3

*Proof.* We prove the suboptimality by constructing a counterexample in which the greedy choice yields a suboptimal matching due to resource contention.

**Setup:** Consider a minimal instance with two workers $\{v_1, v_2\}$ and two tasks $\{t_1, t_2\}$. Set the demand $d = 1.0$ for both tasks, and assume each worker has a remaining capacity of 1. The potential contribution matrix is defined as follows: Worker $v_1$ has a potential of 2.0 for $t_1$ and 1.0 for $t_2$. Worker $v_2$ has a potential of 1.0 for $t_1$ and 0 for $t_2$.

**Greedy Selection:** The greedy algorithm prioritizes the pair with the highest marginal gain. $v_1$ has a raw potential of 2.0 at $t_1$. Although the effective gain is truncated at 1.0 due to demand saturation, standard heuristics prioritize $v_1$ over $v_2$. Consequently, the algorithm selects $(v_1, t_1)$, consuming the capacity of $v_1$. As a result, $t_1$ is satisfied, but $t_2$ remains unsatisfied because the remaining worker $v_2$ lacks the capability to complete it. The total utility achieved is 1.0.

**Optimal Selection:** In contrast, the global optimal strategy reserves the versatile worker $v_1$ for the task only they can complete. By assigning $v_2 \rightarrow t_1$ and $v_1 \rightarrow t_2$, both tasks are fully satisfied, yielding a total utility of 2.0.

**Conclusion:** The greedy strategy achieves only 50% of the optimal utility in this instance, demonstrating that local optimality does not guarantee global optimality under capacity constraints. □

## G. Training Process of Rainbow DQN

The detailed training procedure for our proposed GKD-Recruiter is outlined in Algorithm 1. We employ the Rainbow DQN framework to optimize the seed selection policy effectively.

---

**Algorithm 1:** RAINBOW DQN TRAINING

---

**Input:** Social Graph $\mathcal{G}$, Task Set $\mathcal{T}$, Budget $K$, Max Episodes $E$, Update Frequency $C$
**Output:** Trained Q-network parameters $\theta$

1 Initialize prioritized replay buffer $D$
2 Initialize neural network $Q$ with parameter set $\theta$ and target neural network $Q'$ with parameter set $\theta'$, $\theta = \theta'$

3 **for** *episode* $\leftarrow 1$ *to* $E$ **do**
4     Initialize seed worker-task pair set $S$ as $\emptyset$
5     Initialize the representation vector $x_s$ of seed set $S$
6     **for** $t \leftarrow 1$ *to* $K$ **do**
7         Sample a noisy network $\xi$
8         Select pair $(v_i, t_j)$ by $\arg\max_{x_a} Q(x_s, x_a, \xi, \theta)$
9         Calculate $reward^t$ according to Equation (4.4)
10         $S^{t+1} \leftarrow S^t \cup \{(v_i, t_j)\}$ and calculate $x_s^{t+1}$
11         Store $[x_s^t, x_a^t, reward^t, x_s^{t+1}]$ into $D$
12         Randomly sample a minibatch data from $D$
13         Sample the noisy variable $\xi$ for $Q$
14         Sample the noisy variable $\xi'$ for $Q'$
15         **if** $t$ *is* $K$ **then**
16             $y^t \leftarrow reward^t$
17         **else**
18             Calculate $y^t$
19         Use SGD method to take a gradient descent step on $(y^t - Q(x_s^t, x_a^t, \theta))^2$
20         After $C$ steps, update $\theta' \leftarrow \theta$

---

## H. Inference Efficiency on Small-Scale Dataset

Table 8 presents the inference time comparison on the smaller dataset ($|V| = 3000$). Consistent with the results on the large-scale dataset, GKD-Recruiter demonstrates significant efficiency advantages over simulation-based methods (e.g., CELF, ComGreedy) and achieves comparable speed to other heuristic and RL-based baselines.

*Table 8.* Inference Time (seconds) on Small-scale Dataset ($|V| = 3000$)

| Model \ $|S|$ | 25 | 50 | 75 | 100 | 150 |
|---|---|---|---|---|---|
| GKD-Recruiter | 0.58 | 1.14 | 1.75 | 2.74 | 3.51 |
| DegGreedy | 0.08 | 0.20 | 0.17 | 0.23 | 0.44 |
| NDD | 0.13 | 0.25 | 0.37 | 0.53 | 0.73 |
| FastSelector | 2.81 | 5.38 | 8.14 | 11.91 | 15.61 |
| ComGreedy | 18.00 | 36.72 | 53.88 | 77.70 | 104.61 |
| CELF | 3.74 | 7.22 | 11.31 | 15.56 | 21.82 |
| TSIM | 0.73 | 1.39 | 2.21 | 2.98 | 4.11 |
| MAIM | 1.03 | 1.72 | 2.92 | 4.24 | 6.62 |
| DQNSelector | 1.08 | 1.80 | 2.96 | 4.35 | 6.75 |

## I. Related Topic in ICML

Table 9 summarizes recent ICML publications on crowdsourcing and social network analysis, highlighting the relevance.

*Table 9.* Related ICML Papers on Crowdsourcing and Social Network

| Year | Title | Keywords |
|---|---|---|
| 2025 | TLLC: Transfer Learning-based Label Completion for Crowdsourcing | Crowdsourcing |
| 2025 | From Crowdsourced Data to High-quality Benchmarks: Arena-Hard and Benchbuilder Pipeline | Crowdsourcing |
| 2025 | Label Distribution Propagation-based Label Completion for Crowdsourcing | Crowdsourcing |
| 2024 | Unbiased Multi-Label Learning from Crowdsourced Annotations | Crowdsourcing |
| 2023 | Recovering Top-Two Answers and Confusion Probability in Multi-Choice Crowdsourcing | Crowdsourcing |
| 2021 | Crowdsourcing via Annotator Co-occurrence Imputation and Provable Symmetric Non-negative Matrix Factorization | Crowdsourcing |
| 2025 | SNS-Bench: Defining, Building, and Assessing Capabilities of Large Language Models in Social Networking Services | Social Network |
| 2025 | Optimizing Social Network Interventions via Hypergradient-Based Recommender System Design | Social Network |
| 2023 | Quantifying Human Priors over Social and Navigation Networks | Social Network |

