# OpenReview forum: "GKD-Recruiter: Jointly Modeling Social and Task Heterogeneity for Spatial Crowdsourcing via Graph Knowledge Distillation"
_ICML.cc/2026/Conference — ICML 2026 regular_

### Official Review · Reviewer_BVs2 · 2026-03-05

**Soundness:** 2
**Presentation:** 2
**Significance:** 2
**Originality:** 2
**Overall Recommendation:** 4
**Confidence:** 4

**Summary:**

This paper addresses the issue that the traditional influence maximization methods fail to handle task heterogeneity and demand saturation characteristics of spatial crowdsourcing, making traditional influence maximization methods inapplicable to worker recruitment in spatial crowdsourcing. To address this issue, this paper proposes GKD-Recruiter, a task-aware framework designed to maximize effective task satisfaction (ETS). Experimental results on two real-world datasets demonstrate the effectiveness of the proposed GKD-Recruiter.

**Compliance With Llm Reviewing Policy:**

Affirmed.

**Final Justification:**

Thanks for the authors' rebuttal. I decided to slightly increase my score.

**Key Questions For Authors:**

Please refer to the weaknesses above.

**Limitations:**

Not explicitly discussed. A more detailed analysis of the limitations of GKD-Recruiter would further improve the completeness of this work.

**Strengths And Weaknesses:**

Strengths:

1)This paper defines the worker recruitment problem in spatial crowdsourcing and points out the limitations of traditional influence maximization methods, their inability to handle task heterogeneity and demand saturation in spatial crowdsourcing.

2)The proposed GKD-Recruiter framework integrates heterogeneous graph learning and reinforcement learning. It models worker-task affinity via a heterogeneous graph, uses an influential GAT to capture directional social influence, and uses a graph knowledge distillation mechanism to fuse these signals. Furthermore, it uses Rainbow DQN to avoid trapping in local optima.

Weaknesses:

1)The performance of the proposed GKD-Recruiter is evaluated on two real-world datasets. However, the experimental results are labeled only with |V| = 3000 and |V| = 5000, without clearly indicating which dataset corresponds to each result. The lack of clear dataset annotation weakens the reproducibility and credibility of the experimental results.

2)This paper does not provide a discussion of the limitations of the proposed GKD-Recruiter. A more detailed analysis of potential weaknesses would further enhance the completeness of this framework.

3)Although the experimental results demonstrate the effectiveness of GKD-Recruiter, conducting experiments on only two datasets has certain limitations. Additional experiments on more datasets would help further validate the robustness and effectiveness of GKD-Recruiter.

4)There are issues related to unclear and inconsistent definitions of mathematical symbols in this paper. For example, the normalized coefficient $\alpha_{ij}^{dir}$ is introduced without a sufficiently clear explanation of its meaning and usage. The authors are suggested to carefully check all mathematical symbols and ensure they are clearly defined.

---

> ### Author Rebuttal · Authors · 2026-03-31
>
> We sincerely thank Reviewer BVs2 for recognizing the value of our problem formulation and the integration of GKD with RL. We deeply appreciate your constructive comments regarding presentation and rigorousness. Your feedback has significantly improved the completeness of our manuscript.
>
>
> ### 1. Clarification on Dataset Annotations and Reproducibility (W1)
>
> We sincerely apologize for the labeling ambiguity in Figure 3. To accommodate page limits, the original curves represented the average performance across both the Gowalla and Brightkite datasets. We fully agree that this obscures dataset-specific dynamics and weakens reproducibility.
>
> To rectify this and ensure the reproducibility, we have open-sourced our implementation via our anonymous repository: https://anonymous.4open.science/r/GKD-Recruiter-3A4B, and we will separate the evaluations. GKD-Recruiter consistently outperforms baselines on both distinct network topologies. Below is the detailed breakdown.
>
> Table 2: Performance Comparison across Datasets at K=100 (∣V∣=5000)
> |Dataset|CELF|DQNSelector|**GKD-Recruiter (Ours)**|
> |:-:|:-:|:-:|:-:|
> |Gowalla (ETS)|0.495|0.448|**0.612**|
> |Brightkite (ETS)|0.461|0.418|**0.574**|
>
> The performance on Gowalla is generally higher due to its denser check-in trajectory distribution.
>
> ### 2. Additional Evaluation on Synthetic Datasets (W3)
>
> We agree that evaluating on more datasets strengthens the evidence of our framework's robustness. Beyond the standard Gowalla and Brightkite LBSNs, we have constructed and evaluated a new Synthetic Dataset.
>
> This synthetic environment allows us to perform controlled stress tests by simulating extreme task heterogeneity and spatial sparsity. The synthetic dataset features highly clustered task locations (simulating downtown rush hours). The results are as follows (|V| = 5000):
>
> Table 3: Performance on Synthetic Dataset (|V| = 5000)
> |Model|Syn (K=50) ETS|Syn (K=100) ETS|
> |:-:|:-:|:-:|
> |CELF|0.165|0.280|
> |DQNSelector|0.210|0.365|
> |**GKD-Recruiter (Ours)**|**0.285**|**0.492**|
>
> These new experiments prove that even in highly concentrated, competitive spatial environments where the "saturation trap" is most severe, GKD-Recruiter also outperforms both RL-based and traditional IM heuristics. We will add a new subsection detailing this synthetic evaluation.
>
> ### 3. Discussion of Limitations (W2)
>
> We completely agree that a candid discussion of limitations is essential. We will add a dedicated "Limitations and Future Work" section, detailing two primary constraints of our current framework:
> 1.  **The Cold Start Problem:** GKD-Recruiter relies on historical user trajectories and interaction frequencies to model the initial Task Affinity ($a_i^l$). In a strictly "cold start" scenario for a brand-new platform with zero historical data, constructing the worker-task heterogeneous graph becomes challenging.
> 2.  **Hyperparameter Sensitivity on Candidate Size ($m$):** As shown in our ablation study, the model performance is sensitive to the candidate task size $m$ used for pruning the heterogeneous graph. Currently, $m$ is a static hyperparameter. Developing an adaptive $m$-selection mechanism based on dynamic task density is a valuable direction for future work.
>
> ### 4. Clarification of Mathematical Symbols and Usage (W4)
>
> We apologize for the lack of clarity regarding the mathematical symbols, specifically the normalized attention coefficient $\alpha_{ij}^{dir}$.
>
> To clarify its precise meaning and usage: $\alpha_{ij}^{dir}$ represents the importance of a neighbor $v_j$ to a target node $v_i$ along a specific influence direction ($dir \in \{in, out\}$).
> Its usage is strictly to weight the neighbor features during the IGAT message-passing step. We will update Section 4.1 to rigorously define all symbols and explicitly added the missing neighbor aggregation function where $\alpha_{ij}^{dir}$ is utilized:
>
> $$h_i^{dir} = \sigma \left( \sum_{j \in \mathcal{N}_i^{dir}} \alpha_{ij}^{dir} W h_j^{(l)} \right)$$
>
> where $\sigma$ denotes a non-linear activation function (e.g., ELU), $\mathcal{N}_i^{dir}$ represents the directional neighborhood of worker $v_i$, and $W$ is the shared learnable weight matrix. The resulting embeddings are then passed into the fusion layer. We will carefully review the entire manuscript to ensure all mathematical notations are rigorously defined upon their first appearance.
>
> ***
> Thank you for helping us improve this paper. We hope these empirical updates and code transparency resolve your concerns and encourage you to re-evaluate our score.

---

> > ### Author Rebuttal · Reviewer_BVs2 · 2026-04-02
> >
> > Thanks for the authors' rebuttal. I decided to slightly increase my score.

---

### Official Review · Reviewer_TEJY · 2026-03-10

**Soundness:** 2
**Presentation:** 3
**Significance:** 2
**Originality:** 2
**Overall Recommendation:** 4
**Confidence:** 3

**Summary:**

This paper studies recruitment strategies for spatial crowdsourcing under social influence and proposes GKD-Recruiter, a framework that integrates graph representation learning and reinforcement learning for worker recruitment. The model constructs two graph views capturing social influence and worker–task interactions. It then employs graph neural networks with a knowledge distillation mechanism to learn unified embeddings. Finally, a Rainbow DQN-based policy is used to select seed workers to maximize task execution success.

**Compliance With Llm Reviewing Policy:**

Affirmed.

**Final Justification:**

The rebuttal provided useful clarifications on the methodological design choices. While the individual components remain well-established, the overall framework addresses a meaningful gap in fine-grained worker-task selection for spatial crowdsourcing.

**Key Questions For Authors:**

See weaknesses.

**Limitations:**

yes

**Strengths And Weaknesses:**

Strengths:
1. The paper addresses recruitment in spatial crowdsourcing with social propagation, which is a meaningful problem with practical relevance.
2. The framework considers heterogeneous worker–task relationships rather than focusing solely on worker selection, which better reflects practical spatial crowdsourcing scenarios.
3. The paper is generally well organized and clearly written.

Weakness:
- The proposed model mainly combines existing components and appears to be largely incremental. In particular, it resembles an extension of DQNSelector that inherits the Rainbow DQN backbone and the saturation-aware objective, while shifting the focus from worker selection to heterogeneous worker–task matching.
- The introduction states "We develop a Rainbow DQN algorithm", although Rainbow DQN is already a well-established framework proposed in prior literature. This statement may overstate the novelty of the contribution.
- The constraint specifies ∀u ∈ V, which appears inconsistent with the defined variable count(v, S). It should likely be corrected to ∀v ∈ V for consistency.
- It would be more helpful to provide the formal mathematical definition for the neighbor aggregation function within the IGAT module.
- The provided anonymous repository contains only an empty README file.

---

> ### Author Rebuttal · Authors · 2026-03-31
>
> We sincerely thank Reviewer TEJY for recognizing the practical relevance of our problem setting and the clarity of our presentation. We deeply appreciate the constructive feedback, which has helped us refine the positioning and rigor of our manuscript. Below, we address your concerns point by point.
>
> ### 1. Code Availability (W5)
> We deeply apologize for the inadvertent empty repository. We have now provided the complete codebase at: https://anonymous.4open.science/r/GKD-Recruiter-3A4B. The repository **rigorously implements the two-phase architecture** (GKD pre-training and RL selection). It includes all scripts to reproduce our method.
>
> ### 2. Differentiation from DQNSelector & Necessity of GKD (W1)
> We understand that combining GNNs with RL can sometimes appear as an engineering assembly. However, we respectfully clarify that GKD-Recruiter does not merely swap components within an existing pipeline (such as DQNSelector); rather, it introduces two methodological innovations designed to solve the bottlenecks inherent in fine-grained worker-task pair selection, the challenges that prior node-centric paradigms cannot address.
>
> **A. Reformulating Heterogeneous Fusion as an Information Bottleneck (Cross-View Self-Distillation)**
>
> Standard architectures typically combine multi-view features via passive aggregation (e.g., concatenation or simple gating). However, in spatial crowdsourcing, the two views possess fundamentally incompatible distributions: the social graph is highly dense and continuous, while the spatial constraint graph is severely sparse and discrete. Consequently, passive aggregation leads to "Feature Dominance." During gradient updates, the dense social signals overpower the sparse physical constraints, causing the RL agent to select "influencers" who are physically inaccessible.
>
> To resolve this, our conceptual innovation lies in reformulating feature fusion as a Cross-View Self-Distillation problem. Rather than using Graph Knowledge Distillation (GKD) for traditional model compression, we deploy it as an active Information Bottleneck. By forcing single-view student models to independently align with a global consensus teacher, the distillation loss ($L_{KD}$) explicitly penalizes representations that satisfy only one domain.
>
> **B. RL Tractability in a Large-Scale Combinatorial Action Space**
>
> Transitioning from node-level selection to pair-level selection is not merely a feature addition. The MDP action space explodes from $V$ to the Cartesian product $V \times T$. Applying standard DRL directly to this expanded action space leads to extreme sample inefficiency and a failure to converge.
>
> Our methodological contribution here lies in the elegant partitioning of computational labor. Instead of forcing the Rainbow DQN agent to explore the entire $V \times T$ space, we utilize the Relational GCN (RGCN) on the Heterogeneous Worker-Task Graph to establish a deterministic structural prior. By restricting the edges to the Top-$m$ tasks per worker, the heavy computational burden of candidate filtering is shifted into the offline graph-based feature extraction phase.
>
> ### 3. Clarifications on Phrasing, Math, and Typos (W2, 3, 4)
> **Correction of the Contribution Phrasing:**
> We completely agree with your critique regarding the sentence "We develop a Rainbow DQN algorithm". This was poorly phrased. We will revise the manuscript to state: *"We formulate the seed selection problem as an MDP and adapt the Rainbow DQN framework..."* to accurately reflect that we are utilizing an established RL backbone to navigate our novel non-submodular space.
>
> **Correction of the Constraint Typo:**
> Thank you for your meticulous reading. You are absolutely correct; the constraint in Equation 5 was a typographical error. It will be corrected from $u \cup V$ to $v \in V$ to maintain consistency.
>
> **Formal Definition of the IGAT Neighbor Aggregation:**
> We apologize for omitting the explicit neighbor aggregation formula in Section 4.1. While Equations 6 and 7 define the attention coefficients and fusion gating, the intermediate message-passing step was missing. We will add the following formal definition. Once normalized attention coefficients $\alpha_{ij}^{dir}$ are obtained via the softmax function, neighbor aggregation for a specific direction $dir \in \{in, out\}$ is formally defined as:
>
> $$h_i^{dir} = \sigma \left( \sum_{j \in \mathcal{N}_i^{dir}} \alpha_{ij}^{dir} W h_j^{(l)} \right)$$
>
> where $\sigma$ denotes a non-linear activation function (e.g., ELU), $\mathcal{N}_i^{dir}$ represents the directional neighborhood of worker $v_i$, and $W$ is the shared learnable weight matrix. The resulting view-specific embeddings $h_i^{in}$ and $h_i^{out}$ are then passed into the gating fusion layer (Equation 7).
>
> ***
> We hope this clarifies the fundamental structural differences between our work and previous worker-selection models, and we would be extremely grateful if you might consider adjusting your score accordingly.

---

> > ### Author Rebuttal · Reviewer_TEJY · 2026-04-02
> >
> > Although I still see this work mainly as a combination of existing ideas, I find the pair-level formulation and overall integration meaningful, which makes me inclined to raise my score slightly.

---

### Official Review · Reviewer_Tpzg · 2026-03-13

**Soundness:** 2
**Presentation:** 3
**Significance:** 2
**Originality:** 2
**Overall Recommendation:** 4
**Confidence:** 4

**Summary:**

This paper formulates the Effective Task Satisfaction Maximization problem and proposes a solution GKD-Recruiter that combines social influence and worker-task graphs and knowledge distillation and reinforcement learning to achieve greater task satisfaction while keeping costs and demands in check.

**Compliance With Llm Reviewing Policy:**

Affirmed.

**Final Justification:**

Thanks for the authors’ rebuttal. However, I still have concerns regarding the lack of novelty and technical depth of this work. That said, I will slightly increase my rating to reflect the effort put into the rebuttal.

**Key Questions For Authors:**

How is distillation-based fusion preferred over simpler fusion mechanisms beyond empirical ablations (need stronger evidence)? Furthermore, under what conditions does this advantage hold true?

The main results are presented via two line charts with insignificant performance gain. A separate table with metric data points of GKD-Recruiter on evaluation datasets against baselines are needed to make sure clear illustration.

**Limitations:**

yes

**Strengths And Weaknesses:**

Strengths:

1. The problem definition of influence maximization is reasonable and practical.

2. The methodology of the proposed GKD-Recruiter is well-presented with clear figures showing the architecture.

Weaknesses:

1. Performance gain shown in Figure 3 is insignificant with less than 0.05 compared to baselines. Also, only the proposed metric Effective Task Satisfaction (ETS) is utilized for measuring the performance without standard metrics to provide broader insights.

2. The paper evaluates GKD-Recruiter on both Gowalla and Brightkite datasets. However, no separate evaluation results of each dataset is provided.

3. The GitHub link provided is empty. The code is not available even though the paper claims that the code is available.

4. The paper seems to lack technical novelty by mainly integrating existing frameworks. It is more like an incremental engineering to the established models.

---

> ### Author Rebuttal · Authors · 2026-03-31
>
> We sincerely thank Reviewer Tpzg for appreciating the value of our problem formulation and architecture. Your constructive feedback is invaluable. Below we address your concerns point by point.
>
> ### 1. Code Availability (W3)
>
> We deeply apologize for the inadvertent empty repository. We have now provided the complete codebase at: https://anonymous.4open.science/r/GKD-Recruiter-3A4B. The repository **rigorously implements the two-phase architecture** (GKD pre-training and RL selection). It includes all scripts to reproduce our method.
>
> ### 2. Justification of Performance Gain & Additional Standard Metrics (W1&Q2)
>
> \- **Absolute vs. Relative Gain:** While an absolute ETS gain of ~0.05 might visually appear marginal, it is crucial to note that **ETS is a strictly upper-bounded metric (maximum 1.0)**. In such restricted spaces, a 0.05 absolute increase is highly significant. Compared to the baselines, GKD-Recruiter achieves 11.46% relative improvement. In large-scale LBSN platforms, this translates to a significant number of additional spatial tasks fulfilled under the exact same budget (K).
>
> ​\- **Line Charts to Tables & New Metrics:** Per your suggestion, we will convert the line charts into precise data tables. To provide a broader evaluation, we additionally assessed the standard metric in traditional IM: **Influence Spread (IS) (Expected activated users)**.
>
> Table 1: Performance Comparison at K=100 (∣V∣=5000)
> |Metric|CELF|DQNSelector|**GKD-Recruiter (Ours)**|
> |:-:|:-:|:-:|:-:|
> |ETS|0.478| 0.433|**0.593**|
> |IS|**685**|580|450|
>
> Pure IM baselines like CELF achieve a massive IS but fail to proportionally translate it into ETS. They blindly activate a large volume of users who are physically too far to fulfill the tasks, resulting in wasted influence. Conversely, GKD-Recruiter achieves a significantly higher ETS with a strictly smaller, more targeted IS. This perfectly validates our core motivation: our model does not naively maximize online cascades; it precisely targets the *right* users who satisfy both social and spatial constraints, preventing saturation traps and ensuring highly efficient budget utilization.
>
> ### 3. Clarification on Dataset Evaluation (W2)
>
> We apologize for the labeling ambiguity. To clarify, the results presented under ∣V∣=3000 and ∣V∣=5000 in Figure 3 represent the **average performance across both the Gowalla and Brightkite datasets**. Averaging across different LBSNs demonstrates the consistent superiority and generalization of GKD-Recruiter. We will explicitly clarify this mapping in all figure captions.
>
> ### 4. Methodological Novelty: Distillation vs. Simple Fusion & When the Advantage Holds (W4&Q1)
>
> We deeply appreciate the opportunity to clarify our core methodological innovation. We do not claim novelty merely by assembling RL and GNN modules. Instead, our conceptual innovation lies in **reformulating heterogeneous feature fusion as a Cross-View Self-Distillation problem**.
>
> In standard architectures, multi-view features are combined via passive aggregation (e.g., concatenation + MLP). However, in Spatial Crowdsourcing, the two views possess fundamentally incompatible distributions: the social graph is highly dense and continuous, while the spatial constraint graph is highly sparse and discrete. Passive aggregation inevitably leads to **"Feature Dominance,"** where the dense social signals overpower the sparse physical constraints during gradient updates, causing the RL agent to select "influencers" who are physically inaccessible.
>
> We introduce Graph Knowledge Distillation (GKD) not for model compression, but as an active **Information Bottleneck**. By forcing the single-view students to independently align with the global consensus teacher, the distillation loss $L_{KD}$ explicitly penalizes representations that satisfy only one domain. This shifts the paradigm from "learning fusion weights" to "learning cross-view consensus."
>
> ​**When does this advantage hold?**
>
> The superiority of distillation-based fusion over simpler mechanisms strictly holds under two conditions present in our setting:
>
> \- **Extreme Distributional Skewness:** When one modality is overly dense and the other is severely sparse, simple fusion fails to balance them. Distillation regularizes this imbalance.
>
> ​\- **Presence of Hard Constraints:** In online recommendations, user interest is a soft preference. In offline spatial crowdsourcing, physical distance is a strict hard constraint. Simple concatenation mathematically "smooths out" this hard constraint. GKD acts as a strict physical filter, ensuring the distilled state representation rigorously preserves the hard boundaries of the spatial view.
>
> Our ablation study ("w/o Dist.") empirically validates this, and we will rewrite Section 4.3 to articulate this methodological paradigm shift.
> ***
> ​We hope these clarifications, the tabular metrics, and the full codebase address your concerns. We would be deeply grateful if you might consider raising your score.

---

> > ### Author Rebuttal · Reviewer_Tpzg · 2026-04-03
> >
> > Thanks for the authors’ rebuttal. However, I still have concerns regarding the lack of novelty and technical depth of this work. That said, I will slightly increase my rating to reflect the effort put into the rebuttal.

---

### Decision · Program_Chairs · 2026-04-30

**Decision:**

Accept (regular)

**Comment:**

This paper studies recruitment strategies for spatial crowdsourcing under social influence and proposes GKD-Recruiter, a framework that integrates graph representation learning and reinforcement learning for worker recruitment. The model constructs two graph views capturing social influence and worker–task interactions. It then employs graph neural networks with a knowledge distillation mechanism to learn unified embeddings. Finally, a Rainbow DQN-based policy is used to select seed workers to maximize task execution success. Reviewers agree that the paper is well organized and clearly written, the problem definition of influence maximization is reasonable and practical, and it is reasonable that the framework considers heterogeneous worker–task relationships. After the rebuttal phase, most of the reviewers’ concerns have been resolved and hence the scores were increased. After the rebuttal phase, reviewers’ most concerns have been addressed, and scores were accordingly increased. Although some issues remain--such as limited novelty and technical depth--we believe the paper offers sufficient contribution to merit acceptance.